# The Presence of Aflatoxin M_1_ in Milk and Milk Products in Bangladesh

**DOI:** 10.3390/toxins13070440

**Published:** 2021-06-25

**Authors:** Abu Hasan Sumon, Farjana Islam, Nayan Chandra Mohanto, Rahanuma Raihanu Kathak, Noyan Hossain Molla, Sohel Rana, Gisela H. Degen, Nurshad Ali

**Affiliations:** 1Department of Biochemistry and Molecular Biology, Shahjalal University of Science and Technology, Sylhet 3114, Bangladesh; hasansumon008@gmail.com (A.H.S.); farjana-bmb@sust.edu (F.I.); ncmohanto@gmail.com (N.C.M.); rahanumaraihanu@gmail.com (R.R.K.); nayanmuntasir@gmail.com (N.H.M.); 2Department of Veterinary and Animal Science, Rajshahi University, Rajshahi 6205, Bangladesh; srana31373@gmail.com; 3Leibniz-Research Centre for Working Environment and Human Factors (IfADo) at the TU Dortmund, Ardeystr. 67, D-44139 Dortmund, Germany; degen@ifado.de

**Keywords:** aflatoxin M_1_, Bangladesh, ELISA, milk, milk products

## Abstract

As milk provides both micro- and macronutrients, it is an important component in the diet. However, the presence of aflatoxin B_1_ (AFB_1_) in the feed of dairy cattle results in contamination of milk and dairy products with aflatoxin M_1_ (AFM_1_), a toxic metabolite of the carcinogenic mycotoxin. With the aim to determine AFM_1_ concentrations in milk and milk products consumed in Bangladesh, in total, 145 samples were collected in four divisional regions (Sylhet, Dhaka, Chittagong, and Rajshahi). The samples comprised these categories: raw milk (n = 105), pasteurized milk (n = 15), ultra-high temperature (UHT)-treated milk (n = 15), fermented milk products such as yogurt (n = 5), and milk powder (n = 5). AFM_1_ levels in these samples were determined through competitive enzyme-linked immunosorbent assay (ELISA). Overall, AFM_1_ was present in 78.6% of milk and milk products in the range of 5.0 to 198.7 ng/L. AFM_1_ was detected in 71.4% of raw milk (mean 41.1, range 5.0–198.7 ng/L), and in all pasteurized milk (mean 106, range 17.2–187.7 ng/L) and UHT milk (mean 73, range 12.2–146.9 ng/L) samples. Lower AFM_1_ levels were found in yogurt (mean 16.9, range 8.3–41.1 ng/L) and milk powder samples (mean 6.6, range 5.9–7.0 ng/L). About one-third of the raw, pasteurized, and UHT milk samples exceeded the EU regulatory limit (50 ng/L) for AFM_1_ in milk, while AFM_1_ levels in yogurt and milk powder samples were well below this limit. Regarding regions, lower AFM_1_ contamination was observed in Chittagong (mean 6.6, max 10.6 ng/L), compared to Sylhet (mean 53.7, max 198.7 ng/L), Dhaka (mean 37.8, max 97.2 ng/L), and Rajshahi (mean 34.8, max 131.4 ng/L). Yet, no significant difference was observed in AFM_1_ levels between summer and winter season. In conclusion, the observed frequency and levels of aflatoxin contamination raise concern and must encourage further monitoring of AFM_1_ in milk and milk products in Bangladesh.

## 1. Introduction

Milk is an important food as it provides micro- and macronutrients essential for the growth and maintenance of human health [1]. As milk and milk products are consumed by all age groups, including young children, dairy milk must be free of toxic compounds, including mycotoxins [2]. However, milk and milk-based products can contain aflatoxin M_1_ (AFM_1_), a metabolite of the mycotoxin aflatoxin B_1_ (AFB_1_), a potent human carcinogen [3]. AFM_1_ occurs in the milk of dairy cattle ingesting feed contaminated with AFB_1_, which is partly converted to this hydroxylated metabolite and then excreted in milk. The fraction of AFB_1_ in feed that is transferred to milk as AFM_1_ (carry-over) ranges between 0.6% and 3% with up to 6.2% in high yielding cows [4,5,6]. Due to its strong toxicity in many species, AFB_1_ is an undesirable substance in animal feed [7], and its levels in feed for dairy cattle are restricted in several countries in order to minimize carry-over and thereby human AFM_1_ exposure with milk [1,8]. 

The presence of AFM_1_ in milk and dairy products represents a worldwide concern for several reasons: (i) although AFM_1_ is known to exert lower carcinogenic potency than AFB_1_, their acute toxicities are rather similar [1,9,10]; (ii) AFM_1_ is heat stable, and normal processing and storage are not effective in reducing its levels in milk and milk products [11,12]; and (iii) small levels of this contaminant may impose health risks for consumers of large quantities of milk products, such as children, a particularly vulnerable subgroup in the population [13,14]. 

To protect humans against adverse effects from mycotoxin exposure, regulatory standards (limit values) for aflatoxins as contaminants in food and feed have been set in many countries. For instance, the European Commission (EC) established a maximum limit (ML) of 50 ng of AFM_1_/kg in milk, and a limit for AFM_1_ of 25 ng/kg in infant formula and follow-up formula [15], while other regulatory authorities and the Codex Alimentarius Commission have set a higher limit of 500 ng/kg in raw milk [16]. 

Aflatoxin contamination of feed commodities is frequent in some regions of the world, including South and Southeast Asia [17,18]. Bangladesh has a tropical monsoon climate, with clear seasonal variations in rainfall, temperature, and humidity; agriculture is highly dependent on the weather, with crop production mainly in summer and winter [19,20]. As high humidity at harvest and in storage favor the growth of aflatoxin-producing fungi, it is expected that crops in Bangladesh are also affected. Indeed, when aflatoxins were analyzed in cereal samples collected in all Bangladeshi districts and six times during a year, maize had the highest prevalence and level of contamination; contaminant levels were lower, yet still significant, in rice and in wheat, and all samples showed considerable seasonal variability [21]. Another study in Bangladesh reported the presence of AFB_1_ in pooled samples of commonly consumed food commodities and in poultry feed [22]. However, data on mycotoxins are scarce, in particular in feeds for livestock, a major part of the economy of Bangladesh [23]. 

For developing countries, where regulation on food and feed contaminants is not in place or not enforced by regular surveillance, monitoring of AFM_1_ occurrence in milk is recommended, as it is easy to perform and an indicator for dairy feed contamination. Furthermore, as milk and its products are important components of the human diet, the presence of AFM_1_ in these products may impose health risks for the consumers. This view is supported by a recent, first study on AFM_1_ contamination of raw milk and processed milk samples, collected at regional small farms and grocery stores in the Dhaka region [24]: AFM_1_ was detected in more than half (53/100) of the samples, and among these, 75% were above regulatory limits set in Europe for AFM_1_ in milk, and 43% exceeded the limit set for AFM_1_ in dairy milk of 500 ng/kg in Bangladesh [25]. The results of this study in the Dhaka region [24] warrant further analysis, including other regions of the country. Furthermore, the frequent presence of AFM_1_ in the urine of adults and children in Bangladesh [20,26,27] indicates the necessity of monitoring and screening food items that may be possible sources of aflatoxin exposure in the Bangladeshi population. Thus, the objective of our present study was to determine AFM_1_ concentrations in milk and milk products collected from four major divisional regions in Bangladesh, and to explore some variables which may affect contamination levels in raw milk. 

## 2. Results

### 2.1. Levels of AFM_1_ in Milk and Milk Products 

The detection frequencies and contamination levels of AFM_1_ in milk (raw, pasteurized, UHT) and in dairy products (milk powder, yogurt) are presented in Table 1 Overall, AFM_1_ was detected in 78.6% of all milk and milk products in the range from 5.0 to 198.7 ng/L (mean 51.5 ng/L). AFM_1_ was detected in 71.4% of raw milk samples (mean 41.1 ng/L, range 5.0–198.7 ng/L), and in 100% of pasteurized milk (mean 106 ng/L, range 17.2–187.7 ng/L) and UHT milk (mean 73 ng/L, range 12.2–146.9 ng/L). AFM_1_ contamination levels were comparatively lower in yogurt (mean 16.9 ng/L, range 8.3–41.1 ng/L) and milk powder samples (mean 6.6 ng/L, range 5.9–7.0 ng/L). In total, 32.4% of milk samples (raw milk 23.8%, pasteurized and UHT milk 73.3%) exceeded the EU regulatory limit (50 ng/kg) for AFM_1_ in milk. AFM_1_ contamination levels in all yogurt and milk powder samples were well below the EU regulatory limit. 

### 2.2. Regional and Seasonal Variations in AFM_1_ Contamination Levels

AFM_1_ contamination levels were compared for the raw milk samples collected from the four divisional regions of Bangladesh. The lowest AFM_1_ contamination was observed in the Chittagong division (mean 6.6 ng/L, max 10.6 ng/L) compared to Sylhet (mean 53.7 ng/L, max 198.7 ng/L), Dhaka (mean 37.8 ng/L, max 97.2 ng/L), and Rajshahi (mean 34.8, max 131.4 ng/L) division (Table 2). The Sylhet region had the maximum amount of milk samples (32.5%) exceeding the EU regulatory limit, whereas all the samples from the Chittagong division were well below this limit. 

Regarding seasonal variations, there was no significant difference at the level of AFM_1_ contamination between samples collected in the winter (mean 39.9 ng/L, max 115.5 ng/L) or summer (mean 41.6 ng/L, max 198.7 ng/L) period (Table 3). Yet, the percentage of milk samples that exceeded the EU regulatory limit was slightly higher in the summer (26.2%) than in the winter (20.5%) season. 

## 3. Discussion

As AFM_1_ may pose a health risk for consumers of milk from dairy cattle fed with AFB_1_-contaminated feedstuffs, regulatory authorities have established limits for this mycotoxin metabolite (see Introduction). Regular monitoring of AFM_1_ is needed to control for compliance with food standards and protecting the population against hazardous dietary exposures. The prevalence and levels of AFM_1_ in milk and dairy products analyzed in various regions of the world (data reviewed by [1,8,28,29]) indicate a wide spectrum of human exposure to AFM_1_, with considerable differences between climate zones and countries. In the context of present work, there are studies on AFM_1_ occurrence in milk and milk products from South-Asian countries (listed in Table 4), which include recent ones from Bangladesh.

Regulation on mycotoxin contamination of major food categories, and limits for AFM_1_ in milk, were issued in 2017 in Bangladesh [25], yet data from related monitoring are recently emerging. The new data presented here on AFM_1_ in milk and milk products show a high prevalence of this contaminant (78.6%) in all samples collected in the four divisional regions in Bangladesh (Table 1). Detection frequency and mean levels of AFM_1_ in pasteurized and UHT milk were higher than those in raw milk, yet the concentration ranges observed in these sample types (5–198 ng/L) were similar. This observation is in accordance with findings that heat treatments in pasteurization and sterilization do not cause notable changes in AFM_1_ content in such products [1,8]. On the other hand, AFM_1_ contamination appeared to be lower in milk powder and yogurt (range 5.9–41.1 ng/L), but the sample numbers analyzed here preclude further conclusions on the effects of processing, although it has been found that AFM_1_ levels can decrease in yogurt production [14,45]. 

The higher number of raw milk samples allowed us to explore possible regional differences in AFM_1_ occurrence. The lowest AFM_1_ prevalence and levels were observed in milk from the Chittagong division (Table 2) with none of the samples exceeding the EU regulatory limit of 50 ng/kg. In contrast, in the Sylhet, Dhaka, and Rajshahi divisions, 32.5%, 28%, and 25%, respectively, of the collected milk had AFM_1_ levels exceeding this standard. Cattle feeding and grazing practices can vary between small farmers and commercial dairy farms. Small farmers are accustomed to open grazing practices of their cattle. In addition, they use stored feed such as dry rice paddy straw, rice husk, and wheat bran. Larger farms usually fully depend on stored feed such as dry rice paddy straw, rice husk, wheat bran, maize, different types of pulses, mustard oil cake, soybean bush, and, to a lesser extent, grass. Due to limited resources for this study, we could not collect background information on farming practices and dairy feeds (green fodder, dry fodder, and concentrate) provided at the different locations. Thus, we presently have no information on differences in AFM_1_ contamination. Yet, plans exist to investigate this aspect further, and to raise awareness on critical aspects such as proper storage of dairy feed and other relevant aspects in mycotoxin contamination [1,31].

Regarding seasonal variations, there was no significant difference in the levels of AFM_1_ contamination between the winter and summer period (Table 3), although the percentage of raw milk samples that exceeded the EU regulatory limit was slightly higher in the summer (26.2%) than in the winter (20.5%) season. Our findings differ from studies in Pakistan and India, at which the AFM_1_ levels in milk were often higher in the rainy/monsoon seasons and lower in the summer [31,36,46,47]. These outcomes are related to various factors, namely environmental conditions (high humidity) conducive for growth of *Aspergillus* in stored feed, and dairy animals getting more compound-feed in winter or more out-pasturing in summer. Yet, in these studies [31,36,46,47], the mean AFM_1_ milk levels in both seasons were high compared to those found in our study in Bangladesh.

Recently, Tarannum et al. [24] analyzed samples collected in the Dhaka district of Bangladesh and reported higher ranges and mean AFM_1_ concentrations in raw milk, while processed (pasteurized, UHT) milk had similar levels of contamination as observed in the present study (Table 4). Thus, both studies document frequent contamination of dairy milk in Bangladesh. Variations in AFM_1_ milk levels have been observed in other countries where multiple studies of this type have been conducted, as in Pakistan and India (Table 4). Such differences observed in the prevalence of AFM_1_ levels may depend on several factors, such as environmental conditions, different farming and feeding practices, and the quality and safety control system of the food business operators concordant with the legislations in force [1,31,36].

While the studies listed in Table 4 differ in milk types and dairy products covered, sample sizes, and methods used for AFM_1_ analysis, the reported outcomes show the frequent occurrence of the mycotoxin metabolite in food intended for human consumption in South Asia. AFM_1_ levels in a high percentage of samples were found to exceed the limit of 50 ng/kg set in the EU, and many samples crossed the 10-fold higher limit permitted in other regions of the world. In particular, the high AFM_1_ levels detected in infant formula in two studies in India [30,32] and in sweets in Pakistan [42] raise concerns. Thus, as these products are consumed by vulnerable parts of the population, they should be included in future sampling plans for AFM_1_ analysis in Bangladesh. AFM_1_ levels reported in dairy milk are higher than those determined recently in human breast milk (51.6%, mean 4.42 ng/L) in Bangladesh [48]. Thus, one can conclude that human breast milk is better and safer for infants than dairy milk. 

In summary, the present study along with study [24] document the frequent presence of AFM_1_ in milk and milk products in Bangladesh, and at levels which often exceed regulatory standards for this toxic contaminant. Therefore, and to protect consumers against potential health risks from exposure to AFM_1_, more extensive and periodic control of AFM_1_ concentration in milk and dairy products is needed. AFM_1_ monitoring is relatively easy to perform, in contrast to an analysis of AFB_1_ contamination in animal feeds due to the heterogeneous distribution of the mycotoxin in raw materials and problems derived from sampling procedures [6]. As AFB_1_ contaminated feed is the source of AFM_1_ in milk, improved feed practices and proper storage conditions must be implemented to keep mycotoxin contamination in animal feedstuffs as low as possible. 

## 4. Conclusions

AFM_1_ was frequently detected in milk and milk products collected in four regions of Bangladesh, and at levels that raise concern for the health of consumers, in particular for young children. Regular surveillance and monitoring are needed to prevent and control aflatoxin contamination in milk and milk products in Bangladesh. Furthermore, government agencies should train farmers by raising awareness of the toxicity of aflatoxins, of proper storage conditions of cattle feed to prevent mold growth, and encouraging further studies at dairy companies in order to reduce potential health risks and economic losses. 

## 5. Materials and Methods

### 5.1. Study Areas and Sample Collection

This study was conducted at the Department of Biochemistry and Molecular Biology, Shahjalal University of Science and Technology, Sylhet, Bangladesh. Between December 2018 and November 2019, a total of 145 cow milk and milk products were collected: raw milk samples were collected from four major divisional regions (Sylhet, Dhaka, Chittagong, and Rajshahi) of Bangladesh, whereas milk products were collected from the Sylhet and Dhaka regions. The samples were purchased to cover the following categories: raw milk (n = 105), fresh pasteurized milk (n = 15), ultra-high temperature (UHT)-treated milk (n = 15), fermented milk products such as yogurt (n = 5), and milk powder (n = 5). Raw milk samples were collected directly from farmers’ houses and dairy farms. Pasteurized milk, UHT-treated milk, and fermented milk products were bought from local retail shops; individual production and expiration dates, if provided, were recorded for the collected samples. Raw milk samples were collected in two seasonal periods (summer: March–October and winter: November–February) to check for variations in AFM_1_ contamination. Each category of milk samples was purchased in units of at least 250 mL. The collected samples were stored at −20 °C and analyzed within 2 months of collection. 

### 5.2. Sample Preparation 

Aflatoxin M_1_ concentration in milk and milk products was measured using a commercial enzyme-linked immunosorbent assay (ELISA) (Helica Biosystems Inc., Santa Ana, CA, USA, catalogue no. 961AFLM01M-96). The sample preparation was performed following the manufacturer’s instructions. Briefly, raw milk samples were placed at a refrigerated temperature overnight to initiate the coagulation of fat molecules. Pasteurized milk, UHT milk, and yogurt samples were refrigerated for 1–2 h. Then, all samples were centrifuged at 4000 rpm for 10 min at room temperature to induce separation of the upper fatty layer. For milk powder, about 10 g of sample was dissolved in 100 mL of distilled water and stirred with a magnetic stirrer for 5 min and centrifuged to separate the fat layer. The upper fatty layer was removed by aspiration, and the lower plasma layer of the milk was used in the assay. 

### 5.3. Laboratory Analyses

Standard solutions and prepared samples (200 μL) were added to the precoated ELISA plates in duplicates and incubated for 2 h at ambient temperature. At the end of incubation, the contents of the wells were discarded, and the wells were washed three times with the washing buffer provided with the assay kit. After the washing steps, 100 μL of the conjugate was added to the wells and incubated for 15 min at room temperature. After incubation, the wells were washed and 100 μL of enzyme substrate was added to each well and incubated for 15 min. Following that, 100 μL of stop solution was added to each well and gently mixed. The absorbance of each microwell was measured at 450 nm by using an ELISA reader (Apollo 11 LB 913, Berthold, Germany) within 15 min. AFM_1_ concentration in each well was calculated using a semi-logarithmic standard curve (prepared using 0, 5, 10, 25, 50, and 100 pg/mL of AFM_1_ solutions), and the mean of the duplicates was used as the final result. To test the accuracy of the AFM_1_ estimations, recovery studies were performed by spiking skim milk samples with three different concentrations of AFM_1_ (5, 10, and 25 pg/mL). The repeatability at these three spike concentrations showed acceptable precisions for AFM_1_ measurements (Table 5). The recovery was 102%, 98%, and 94.8% in the spiked concentration, respectively. The detection limit (LOD) of this ELISA method was 5 pg/mL or 5 ng/L. Samples exceeding the signal for the highest AFM_1_ standard concentration (100 pg/mL) were further diluted and re-tested. 

### 5.4. Statistical Analysis

All data were analyzed statistically using IBM SPSS Statistics version 23. Descriptive analysis was conducted to determine mean, median, and interquartile ranges of the analyte. The obtained data are presented as mean ± standard deviation, and ranges, frequency, and percentiles for the parameters. Differences in AFM_1_ concentrations between seasons and regions were analyzed by an independent sample t-test. One-way ANOVA was used to compare AFM_1_ concentrations in the four divisional regions in Bangladesh. A level of *p*-value < 0.05 was considered statistically significant. 

## Figures and Tables

**Table 1 toxins-13-00440-t001:** Occurrence and levels of AFM_1_ in milk and milk products in Bangladesh.

Samples	N	Positive Samples, n (%)	Mean ± SD (ng/L)	Median (Range) (ng/L)	Above 50 ng/L n (%)
Raw milk	105	75 (71.4)	41.1 ± 43.4 ^a^	23.5 (5.0–198.7)	25 (23.8)
Pasteurized milk	15	15 (100)	106.0 ± 58.5 ^b^	115.7 (17.2–187.7)	11 (73.3)
UHT milk	15	15 (100)	73.0 ± 42.0 ^c^	74.0 (12.2–146.9)	11 (73.3)
Milk powder	5	4 (80)	6.6 ± 0.5	6.7 (5.9–7.0)	0
Yogurt	5	5 (100)	16.9 ± 13.6	12.1 (8.3–41.1)	0
Total	145	114 (78.6)	51.5 ± 50.4	27.9 (5.0–198.7)	47 (32.4)

Only positive samples (≥LOD; 5 ng/L) were considered during mean and median calculation. ^a^ When raw milk is compared to milk powder (*p <* 0.001), ^b^ when pasteurized milk is compared to raw milk (*p <* 0.01), yogurt, and milk powder (*p <* 0.001), ^c^ when UHT milk is compared to yogurt (*p <* 0.01) and milk powder (*p <* 0.001). *p*-values were obtained from one-way ANOVA.

**Table 2 toxins-13-00440-t002:** Levels of AFM_1_ in raw milk samples collected from four divisional regions.

Regions	n	Positive Sample n (%)	Mean ±SD (ng/L)	Median (Range) (ng/L)	Above 50 ng/L n (%)
Sylhet	40	34 (85)	53.7 ± 51.0	34.0 (5.5–198.7)	13 (32.5)
Dhaka	25	15 (60)	37.8 ± 29.2	33.2 (5.1–97.2)	7 (28.0)
Chittagong	20	8 (40)	6.6 ± 1.9 *	5.8 (5.0–10.6)	0 (0)
Rajshahi	20	18 (90)	34.8 ± 39.1	19.2 (6.3–131.4)	5 (25)
Total	105	75 (71.4)	41.0 ± 43.4	23.5 (5.0–198.7)	25 (23.8)

Only positive samples (≥LOD: 5 ng/L) were considered during mean and median calculation. * When comparing mean AFM_1_ concentration to Sylhet (*p <* 0.001), Dhaka (*p <* 0.01) and Rajshahi (*p <* 0.05) regions. *p*-values were obtained from one-way ANOVA.

**Table 3 toxins-13-00440-t003:** Levels of AFM_1_ in raw milk samples collected in winter and summer seasons.

Season	n	Positive Sample n (%)	Mean ± SD (ng/L)	Median (Range) (ng/L)	Above 50 ng/L n (%)
Winter	44	28 (63.6)	39.9 ± 34.3	27.9 (5.5–115.5)	9 (20.5)
Summer	61	47 (77.1)	41.6 ± 43.4	16.5 (5.0–198.7)	16 (26.2)
Total	105	75 (71.4)	41.0 ± 43.4	23.5 (5.0–198.7)	25 (23.8)

Only positive samples (≥LOD: 5 ng/L) were considered during mean and median calculation.

**Table 4 toxins-13-00440-t004:** Occurrence and levels of AFM_1_ in milk and dairy products in South-Asian countries.

Country	Sample Type	Positive Samples n (%)	Mean (ng/L)	Range (ng/L)	Samples ≥ 50 ng/L n (%)	Analytical Method	LOD/LOQ ng/L	References
Bangladesh	Raw Milk	75/105 (71.4)	41.1	5–198.7	25 (23.8)	ELISA (SKM)	5.0/-	Present study
	Pasteurized milk	15/15 (100)	106	17.2–187.7	11 (73.3)			
	UHT milk	15/15 (100)	73	12.2–146.9	11 (73.3)			
	Milk powder	4/5 (80)	6.6	5.9–7.0	0			
	Yogurt	5/5 (100)	16.9	8.3–41.1	0			
Bangladesh	Raw milk	35/50 (70.0)	699.1	22.8–1489.3	34 (97.0)	ELISA (SKM)	18.0/NI	[24]
	Pasteurized milk	13/25 (52.0)	99.8	18.1–672.2	6 (46.0)			
	UHT milk	05/25 (20.0)	35.5	25.1–49.0	0			
India	IMF	17/17 (100.0)	350.0	77.0–844.0	17 (100.0)	ELISA (SKM)	NI	[30]
	Infant formula	17/18 (94.0)	326.0	143.0–770.0	17 (100.0)			
	MBCWF	38/40 (95.0)	267.0	65.0–1012.0	38 (100.0)			
	Liquid milk	4/12 (33.0)	86.0	28.0–164.0	3 (75.0)			
India	Raw milk	110/189 (58.0)	917.0	7.0–13,100.0	96 (50.8)	ELISA (SKM)	5.0/NI	[31]
India	Infant formula	18/18 (100.0)	NI	150.0–713.0	18 (100.0)	ELISA (SKM)	5.0/NI	[32]
	Liquid milk	54/54 (100.0)	NI	172.0–820.0	54 (100.0)			
India	UHT milk	30/45 (66.6)	NI	NI	29 (96.7)	HPLC (IAC)	4.0/NI	[33]
	Raw milk	3/7 (42.9)	NI	NI	NI			
	Pasteurized milk	29/45 (42.9)	NI	NI	3 (10.3)			
India	Buffalo milk	58/150 (38.6)	26.0	NI	9 (16.0)	ELISA (SKM) HPLC (IAC)	5.0/NI (ELISA)	[34]
	Cow milk	68/150 (45.3)	18.0	NI	30 (44.0)			
	Goat milk	50/150 (33.3)	14.0	NI	5 (10.0)			
	Sheep milk	55/150 (36.6)	17.0	NI	7 (12.0)			
	Cheese	2/2 (100.0)	NI	NI	0			
Nepal	Raw milk	8/16 (50.0)	30.0	26.0–138.0	4 (25.0)	TLC	NI	[35]
	Pasteurized milk	6/16 (37.5)	22.0	25.0–127.0	3 (18.8)			
Pakistan	Raw milk	168/168 (100)	199–503	10–700	167 (99.4)	Fluorometric (IAC)	NI	[36]
Pakistan	UHT milk	85 (NI)	254.9	ND-1536.0	80 (94.1)	ELISA (SKM)	4.4/NI	[37]
	Pasteurized milk	78/78 (100.0)	939.5	32.8–4808.0	74 (95.0)			
	Raw milk	55/55 (100.0)	1535.0	1912.0–7460.7	55 (100.0)			
Pakistan	Milk	143/156 (91.7)	346.2	20.0–3090.0	125 (80.1)	ELISA (SKM)	5.0/NI	[38]
Pakistan	Raw milk	137/150 (91.3)	198.8	6.0–554.0	108 (72.0)	ELISA (SKM)	4.3/5.0	[39]
	TW	30/30 (100.0)	113.0	13.0–257.0	17 (56.0)			
	UHT milk	30/30 (100.0)	164.0	10.0–345.0	20 (66.0)			
Pakistan	Milk	76/107 (71)	150.1	4.0–845.4	44 (58.0)	HPLC (IAC)	4.0/NI	[40]
	Yogurt	59/96 (61.0)	94.4	4.0–615.8	28 (47.0)			
	Butter	33/74 (45.0)	69.7	4.0–413.4	17 (52.0)			
Pakistan	Raw milk	63/107 (59.0)	55.0	<4.0–980.0	38 (35.5)	HPLC-FLD (IAC)	4.0/12.0	[41]
Pakistan	Milk	76/107 (71)	212.2	4–845.4	44 (58)	HPLC-FLD (IAC)	4/NI	[40]
	Yogurt	59/96 (61)	147.1	4–615.8	28 (47)			
	White Cheese	93/119 (78)	189.1	4–595.4	14 (15)			
	Cheese cream	89/150 (59)	172.9	4–456.3	10 (11)			
	Butter	33/74 (45)	156.3	4–413.4	17 (52)			
Pakistan	Shop milk	137/175 (78.0)	176.0	2.0–1600.0	50 (28.6)	ELISA (SKM)	2.0/NI	[42]
	Household milk	25/40 (62.0)	470.0	3.0–1900.0	18 (45.0)			
	Farm milk	15/17 (88.0)	110.0	2.0–794.0	7(41.0)			
	Sweets	134/138 (97.0)	480.0	10.0–1500.0	108 (78.0)		10.0/NI	
Pakistan	Milk	18/21 (85.7)	NI	11.0–14.0	0	ELISA (SKM)	NI	[43]
	Yogurt	6/10 (60.0)	NI	10.0–13.0				
	Butter	4/10 (40.0)	NI	7.0–7.4				
Sri Lanka	Milk	29/87 (33.3)	40.2	13.1–84.5	8 (9.2)	HPLC (IAC)	10.0/NI	[44]

NI: not indicated, UHT: ultra-high temperature, IMF: infant milk food, MBCWF: milk based cereal weaning food, TW: tea whitener, ELISA: enzyme-linked immunosorbent assay, HPLC-FLD: high pressure liquid chromatography with fluorescence detection, SKM: skim milk, IAC: immunoaffinity columns. Note: for cheese, EU limit is 250 ng/kg.

**Table 5 toxins-13-00440-t005:** The assay repeatability for AFM_1_ in spiked milk samples.

Spike Level (ng/L)	Repeats (n)	Mean ± SD (ng/L)	Recovery (%)	RSD (%)
5	5	5.1 ± 0.3	102.0	5.9
10	5	9.8 ± 0.2	98.0	2.4
25	5	23.7 ± 0.6	94.8	2.5

## Data Availability

Data are available from the corresponding authors upon reasonable request.

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
