# Peer review of "The Presence of Aflatoxin M_1_ in Milk and Milk Products in Bangladesh"

_toxins, 2021, doi:10.3390/toxins13070440_

Round 1
Reviewer 1 Report
The results are interesting and paint a significant picture of the AM1 contamination problem.
The concentration of AM\ should be calculated with an HPLC method
Author Response
Response to Reviewer 1
The results are interesting and paint a significant picture of the AM1 contamination problem.
Response: Thank you for the appreciating comment
The concentration of AM1 should be calculated with an HPLC method
Response: Thank you for the suggestion. ELISA may not be the gold standard for analyte quantification, yet is a sensitive and suitable tool for detecting AFM1 in milk samples. Thus, ELISA assays were applied in many milk studies worldwide and in South Asia (see Table 5). In developing countries, with limited access to advanced analytical tools such as HPLC-FLD or LC/MS-MS, applying ELISA is often the first or only option to investigate AFM1 frequency and levels in biological samples. Also, with the commercial ELISA used here, we found a good match when comparing ELISA results to those obtained with HPLC-FD in human urine (Ali et al. 2017; doi: 10.1016/j.ijheh.2016.11.002.
Reviewer 2 Report
The paper is a solid, useful contribution to this journal. Nothing is really wrong, but a few points to consider:
1) You don't clearly state what the new regulatory limit is in Bangladesh for milk.
2) It is worth some discussion on what is the common feeding / grazing practices for dairy cattle in Bangladesh. Where are the cattle getting the aflatoxin? Could this be associated with the regional differences in aflatoxin contamination? Would it be more practical to survey milk or to survey dairy feed?
3) The conclusion doesn't seem as well-thought out as the rest of the paper. Is 'monitoring, training and encouraging' the best answer?
Author Response
Response to Reviewer 2
The paper is a solid, useful contribution to this journal. Nothing is really wrong, but a few points to consider:
Response: We appreciate the comment.
1) You don't clearly state what the new regulatory limit is in Bangladesh for milk.
Response: In Bangladesh, the limit for AFM1 in dairy milk is 500 ng/kg, i.e. the same as proposed in Codex Alimentarius. The value is stated in the last paragraph of the introduction with the appropriate reference [25].
2) It is worth some discussion on what is the common feeding / grazing practices for dairy cattle in Bangladesh. Where are the cattle getting the aflatoxin? Could this be associated with the regional differences in aflatoxin contamination? Would it be more practical to survey milk or to survey dairy feed?
Response: Thank you for the valuable comments. In the discussion, we have now included some general info on feed and feeding practice for dairy cattle in Bangladesh; since cows are exposed to aflatoxins (AFB1) mainly with stored cattle feed. Yet this material is usually not analyzed for mycotoxin presence, and the levels may vary from region to region. As stated before: Due to limited resources for this study, we could not collect background information on farming practices and dairy feeds (green fodder, dry fodder, and concentrate, provided at the different locations. We consider it more practical to monitor AFM1 levels in milk, also since fungal toxin contamination in feed can be rather heterogeneous, and thus requires analysis of large batches of cattle feed. Surveying AFM1 in milk will also provide a clear picture of the safety of the product, and inform whether measures need to be taken to reduce hazardous mycotoxin contamination in cattle feed.
3) The conclusion doesn't seem as well-thought out as the rest of the paper. Is 'monitoring, training and encouraging' the best answer?
Response: Regular surveillance is the key to produce safe and good quality milk. As this depends on farmers and feeding practices, we believe that training and raising awareness on dairy cattle exposure to aflatoxins is another practical step. We have edited the text in the conclusion to capture this.